# Crosstalk between Dysfunctional Mitochondria and Proinflammatory Responses during Viral Infections

**DOI:** 10.3390/ijms25179206

**Published:** 2024-08-24

**Authors:** Zitao Sun, Yanjin Wang, Xin Jin, Su Li, Hua-Ji Qiu

**Affiliations:** 1State Key Laboratory for Animal Disease Control and Prevention, Harbin Veterinary Research Institute, Chinese Academy of Agricultural Sciences, Harbin 150069, China; sunzitao1998@163.com (Z.S.); wangyanjin1996@126.com (Y.W.); 2Agricultural College, Yanbian University, Yanji 133002, China; jinxin@ybu.edu.cn

**Keywords:** mitochondrial dysfunction, mitochondrial damage, proinflammatory responses, innate immunity

## Abstract

Mitochondria play pivotal roles in sustaining various biological functions including energy metabolism, cellular signaling transduction, and innate immune responses. Viruses exploit cellular metabolic synthesis to facilitate viral replication, potentially disrupting mitochondrial functions and subsequently eliciting a cascade of proinflammatory responses in host cells. Additionally, the disruption of mitochondrial membranes is involved in immune regulation. During viral infections, mitochondria orchestrate innate immune responses through the generation of reactive oxygen species (ROS) and the release of mitochondrial DNA, which serves as an effective defense mechanism against virus invasion. The targeting of mitochondrial damage may represent a novel approach to antiviral intervention. This review summarizes the regulatory mechanism underlying proinflammatory response induced by mitochondrial damage during viral infections, providing new insights for antiviral strategies.

## 1. Introduction

Mitochondria are important organelles within eukaryotic cells that are crucial for maintaining normal cell function and metabolism. Mitochondria are the main source of energy within eukaryotic cells, and they can produce adenosine triphosphate (ATP) through oxidative phosphorylation in the respiratory chain, providing energy for the physiological functions and metabolism of cells [1]. In addition, mitochondria also play crucial roles in various biological processes, such as cell apoptosis, calcium homeostasis, redox balance, and lipid metabolism. Therefore, mitochondrial dysfunction is closely associated with the onset and progression of various diseases, including cancer, metabolic, cardiovascular, and neurodegenerative diseases, and inflammatory responses induced by viral infections [2].

Mitochondria maintain a dynamic balance by continuously regulating their morphology and function. Their morphology can be regulated through fusion and fission to adapt to the changes in cellular energy demands. In addition, the mitochondrial endoplasmic reticulum (ER) contact point formed between mitochondria and ER is involved in various cellular biological processes, such as calcium homeostasis, lipid synthesis, and cell apoptosis [3]. However, the stimulation of the external environment or abnormal internal metabolism typically leads to mitochondrial damage, resulting in subsequent dysfunction of mitochondria. Mitochondrial damage can also lead to the disruption of mitochondrial dynamic regulatory mechanisms, thereby disrupting the balance of mitochondrial fusion and fission [4,5]. In addition, mitochondrial damage can also activate NLRP3, promoting the release of proinflammatory cytokines. During this process, many viruses have exploited various infection strategies to evade immune responses.

Given the pivotal role of mitochondria in biological functions, the regulation of mitochondrial damage has emerged as a novel strategy for disease treatment. This review aims to summarize the mitochondrial functions and the damage induced by viral infections, as well as their involvement in proinflammatory responses and innate immune responses. Furthermore, we also discussed the potential applications of mitochondria in targeted therapies for various diseases, providing valuable references to further research on the functions of mitochondria in cellular biology and the development of diseases.

## 2. Mitochondrial Structure and Functions

Mitochondria are pivotal organelles within cells, characterized by intricate structures comprising diverse components. Mitochondria consist of four functional regions from the outside to the inside: outer mitochondrial membrane (OMM), inner membrane space (IMS), mitochondrial inner membrane (IMM), and matrix. Typically, mitochondria are elliptical or cylindrical with 0.75 to 3 μm in length, which are double-membrane structure consisting of a smooth outer membrane and an invaginated inner membrane. Mitochondrial cristae are invaginations of the inner membrane, and it not only serves to augment the surface area of the inner membrane but also accommodate electron carrier proteins and ATP synthase. The proteins of IMM are involved in the formation of the electron transport chain (ETC) that provides energy for oxidative phosphorylation.

Generally, viral infections induce changes in mitochondrial structures. It has been shown that human immunodeficiency virus (HIV), human cytomegalovirus (HCMV), and hepatitis C virus (HCV) can disrupt the double-membrane structure of mitochondria by impairing the inner and outer membranes. The mitochondrial permeability transition pore (mPTP) located at contact sites between the inner and outer membranes comprises the voltage-dependent anion channel (VDAC) on the outer membrane, the cyclophilin D (CypD)-binding protein on the inner membrane, and the adenine nucleotide translocator (ANT) at the interface between the inner and outer membranes. Together, they constitute the mPTP channel, which allows the exchange of small molecules and ions (Figure 1) [6].

It has been shown that ROS-induced mitochondrial membrane damage leads to conformational changes in the VDAC or calcium overload, facilitating the formation of the mPTP, which results in the loss of mitochondrial membrane potential and the release of calcium ions from the matrix [7]. The HIV Vpr protein can directly bind to the ANT, promoting the opening of the mPTP [8]. Additionally, the mitochondrial antiviral-signaling protein (MAVS), located at the OMM, mediates signaling of RIG-I and melanoma differentiation-associated gene 5 (MDA5). The activation of MAVS can synergistically promote the expression of type I IFNs (such as IFN-*β*) in the nucleus by activating the IRF3 and nuclear factor kappa B (NF-*κ*B) signaling pathways [9], thereby triggering downstream antiviral responses.

The IMS separates the inner and outer membranes, while the mitochondrial matrix, a gel-like substance containing mitochondrial genetic material (mtDNA), is also present within mitochondria. The small molecule, approximately 16,569 base pairs (bp), encodes proteins involved in cellular respiration and energy production. The released mitochondrial products exacerbate the innate immune response [10]. mtDNA is a typical signaling molecule that transmits danger signals to pattern recognition receptors (PRRs), triggering proinflammatory responses. Reportedly, the dengue virus induces USP18, which regulates the release of mtDNA into the cytoplasm [11].

## 3. Mitochondrial Damage Induced by Viral Infections

### 3.1. The Increased Mitochondrial Membrane Permeability Induced by Viral Infections Leads to Mitochondrial Damage

Viruses can induce mitochondrial damage through various complex processes, among which virus-induced changes in mitochondrial membrane permeability have been demonstrated to exacerbate mitochondrial injury. Viruses can induce the opening of mPTP and accelerate the leakage of mitochondrial matrix substances, leading to mitochondrial dysfunction (Table 1) [12].

Under physiological conditions, the proapoptotic proteins BAX and BAK of the Bcl-2 family shuttle between the OMM and the cytosol. Upon activation, they accumulate and oligomerize on the MOM, enhancing mitochondrial outer membrane permeability (MOMP) and promoting the release of proapoptotic factors, such as cytochrome C (CytC), into the cytosol, thereby triggering the downstream apoptotic signaling pathway. Porcine deltacoronavirus (PDCoV) infection leads to the BAX-mediated MOMP, inducing the release of mitochondrial CytC into the cytosol [13]. Both herpes simplex virus 1 (HSV-1) and Semliki Forest virus (SFV) can trigger MOMP and downstream apoptosis in host cells through the Puma protein [14].

Mitochondria can also indirectly enhance their membrane permeability via the ER. BOK, a homologous family member of BAX and BAK, is distributed in the ER membrane, Golgi apparatus membrane, mitochondrial outer membrane, and ER–mitochondria contact sites. It can induce MOMP in the absence of BAX and BAK. During viral infection-induced ER stress, MOMP is increased by targeting BOK via the ER-associated degradation (ERAD) E3 ubiquitin ligase gp78 and its associated protein VCP. Mitochondria-associated membranes (MAMs) are contact sites between the ER and mitochondria. During RNA virus infection, the cytosolic RIG-I is recruited to MAMs and binds to the adapter protein MAVS to activate the RIG-I signaling pathway. The formation of the IP3R-GRP75-VDAC1 complex, composed of VDAC1 on the mitochondrial outer membrane, cytosolic glucose-regulated protein 75 (GRP75), and IP3R on the ER, serves as a channel for Ca^2+^ transfer from the ER to mitochondria (Figure 2) [15,25,26,27,28,29,30]. The substantial accumulation of Ca^2+^ in mitochondria induces mitochondrial reactive oxygen species (mROS), leading to mitochondrial protein and lipid damage, disruption of the ETC functions, increased mitochondrial membrane permeability, and thereby promoting the production of proinflammatory cytokines. It has been shown that the interaction among the GP5 protein of porcine reproductive and respiratory syndrome virus (PRRSV) and the ER Ca^2+^ release channel IP3R promotes the oligomerization of the mitochondrial outer membrane channel protein VDAC1, facilitating Ca^2+^ flow from the ER to the mitochondria [15].

Additionally, several viruses can trigger the activation of mitochondrial outer membrane receptors and alter mitochondrial membrane permeability through the release of proinflammatory cytokines during viral infections. Viral infections can promote mtDNA release via the mitochondrial calcium uniporter (MCU). The release of mtDNA in epithelial and myeloid cells may also be triggered by proinflammatory cytokines, such as tumor necrosis factor (TNF) and IL-1*β*. Dengue virus infection stimulates host cells to release IL-1*β*, reducing mitochondrial membrane potential and promoting the release of mtDNA [16]. During infection, ROS within host cells can also promote the release of mtDNA, inducing mitochondrial membrane damage [31].

### 3.2. Viral Proteins-Induced Disruption of Mitochondrial Dynamics Leads to Mitochondrial Damage

Viral infections usually induce changes in mitochondrial dynamics, including fusion, fission, and autophagy. Mitochondrial fusion refers to the process in which two or more mitochondria fuse to form larger mitochondria. OMM fusion occurs before IMM fusion. Mitofusin 1 (MFN1) and MFN2 mediate the fusion of OMM through homotypic and heterotypic interactions driven by the hydrolysis of guanosine-5′-triphosphate (GTP) [32]. Under normal circumstances, mitochondrial fusion facilitates the homogenization and equilibrium of mitochondrial contents, maintaining their integrity and function. However, several viruses can interfere with the balance of mitochondrial fusion, leading to excessive or insufficient fusion of mitochondria, thereby affecting the number and distribution of mitochondria within cells and the subsequent cellular metabolism and energy production. It has been shown that Zika virus infection reduces the level of MFN2 protein and reduces mitochondrial fusion [17]. Furthermore, HIV infection results in a reduction in the Vpr-associated MFN2, leading to a reduction in functional MFN2, which subsequently diminishes the frequency of mitochondrial fusion [18,33].

Mitochondrial fission refers to the process in which mitochondria divide into two or more smaller mitochondria. Similarly, the invasion of viruses can also interfere with the balance of mitochondrial fission. Specifically, the non-structural protein 1 (NS1) of influenza virus can promote mitochondrial fragmentation [19]. In addition, mitochondrial fission is also observed in cases of hepatitis B virus (HBV) and HCV infections, and the viral infections induce the phosphorylation of dynamin-1-like protein (DNM1L) at Ser616, upregulation of the expression of DNM1L and mitochondrial fission factor (MFF), recruitment of DNM1L to the mitochondria, and the subsequent increase in mitochondrial fission, thereby mediating the PRKN-dependent mitophagy [20,21].

The crosstalk between the mitochondrial dynamics and autophagy is essential for the maintenance of mitochondrial homeostasis. Autophagy can selectively remove the damaged or aged mitochondria in host cells. During this process, autophagosomes encapsulate damaged mitochondria and transport them to lysosomes for degradation. Mitochondrial autophagy engages two different regulatory mechanisms. One mechanism is the recruitment and interaction between the key autophagy protein LC3, and receptors anchored on OMM, triggering mitochondrial autophagy. Another is induced by the E3 ubiquitin ligase parkin (PARK2, also known as PRKN) and the protein kinase PINK1 (also known as PARK6). Under normal circumstances, the synergistic effect of mitochondrial fission and autophagy facilitates the release and clearance of damaged mitochondria induced by ROS accumulation, maintaining the quality and function of mitochondria [34,35,36].

Viruses have evolved multiple strategies to trigger and manipulate mitochondrial autophagy, generating a favorable microenvironment for viral replication. Viral infections usually trigger the cellular parkin RBR E3 ubiquitin protein ligase (PRKN)-dependent or receptor-mediated mitochondrial autophagy by inducing mitochondrial dysfunction, while other viruses can induce mitochondrial autophagy through their own viral proteins. Some viruses target mitochondria and disrupt the balance of mitochondrial autophagy, leading to immunoevasion [37]. The expression of the HBV HBx protein inhibits lysosomal acidification to reduce the lysosomal degradative capacity [38,39]. Additionally, HBx disrupts the association of the BECN1-Bcl-2 complex, thereby hindering the assembly of pre-autophagosomal structures. The abnormal mitochondrial autophagy may lead to the aggregation and accumulation of damaged mitochondria, resulting in an increase in intracellular oxidative stress and proinflammatory responses.

Influenza A virus (IAV) infection regulates the activation of the autophagy-related signaling pathways to facilitate viral replication. More specifically, IAV infection results in the reduction of the phosphorylation of mTOR and its downstream substrates, such as the 4E-BP1, p70S6K, and S6 proteins, in a time-dependent manner, indicating that IAV infection induces autophagy [22]. Taken together, viral infections induce changes in mitochondrial fusion, fission, and autophagy, thereby impairing the quantity, morphology, and function of mitochondria.

### 3.3. The Mitochondria-Mediated Stress Response during Viral Infections Leads to Mitochondrial Damage

Mitochondria serve as the primary source of energy for maintaining cellular life functions, predominantly accomplished through electron transfer chains located on the inner membrane of mitochondria. They undergo oxidative phosphorylation to produce ATP for sustained energy supply. During this process, electron leakage occurs, mainly at complexes I and III, where the leaked electrons combine with oxygen molecules to produce ROS. At the same time, there is also a well-established antioxidant system within the cell to maintain stable oxygen content, such as superoxide dismutase (SOD), catalase, peroxidase reductase, and glutathione system, which function together to maintain normal physiological functions of the cells. After virus entry, the metabolic environment of host cells may be reshaped, which may lead to mitochondrial dysfunction and increased ROS, as well as damage to the antioxidant system.

Viral infections usually impair the function of mitochondria through multiple strategies, leading to mitochondrial stress. Viruses can interact with the mitochondrial membrane and induce mitochondrial membrane potential (MMP) depolarization and increased membrane permeability. Reportedly, the HCV E1, E2, and NS3 proteins can inhibit electron transfer from respiratory complex I, resulting in MMP depolarization and ROS production [23]. During HIV infection, the Tat protein is translocated from the nucleus to the mitochondria [24]. Meanwhile, mitochondrial membrane depolarization occurs, leading to the production of ROS. The release of ROS leads to alterations in protein modifications and a reduction in lipid or DNA synthesis [40,41,42,43]. The most typical case, the 7,8-dihydro-8-oxy-deoxyguanosine (8-oxo-dG) formed by ROS, attacks on DNA, which can cause base switching and gene mutations [44]. Mammals have also evolved various defense mechanisms to cope with such damage, such as double-strand break repair, nucleotide cleavage repair, and mismatch repair, which greatly reduce the adverse effects of ROS on mtDNA [45,46,47]. Reportedly, NLRP3 can bind to the oxidized mitochondrial DNA (ox-mtDNA), which is released during cell apoptosis. The activated inflammasomes can also increase the secretion of the ox-mtDNA in vitro.

Host cells can also reduce excessive intracellular ROS levels by regulating the expression of antioxidant enzymes, such as superoxide dismutase 2 (SOD2) and glutathione peroxidase 1 (GPX1). Moreover, the nuclear factor E2-related factor 2 (Nrf2) is also another key transcription factor that regulates oxidative stress [48,49]. Nrf2 is activated upon oxidative stress conditions, leading to its dissociation from Keap1 and the subsequent nuclear translocation, thereby enhancing the expression of antioxidant genes to maintain the homeostasis of intracellular ROS. Finally, it serves as a regulatory mechanism to facilitate proper protein folding through the upregulation of molecular chaperones heat shock protein (HSP) 60 and HSP10, or by employing the caseinolytic protease P (ClpP) for the degradation of misfolded proteins [50,51].

## 4. Proinflammatory Response Induced by Mitochondrial Damage upon Viral Infections

The proinflammatory response represents an immune response in response to the disruption of the host’s homeostasis and the receipt of danger signals, including invasion of exogenous pathogenic microorganisms, self-inflammation, trauma, and metabolic disorders. The response is primarily triggered by pathogen-associated molecular patterns (PAMPs) and damage-associated molecular patterns (DAMPs), which can be recognized by the pattern recognition receptors [52,53].

The proinflammatory response primarily consists of two steps: the activation of the NF-*κ*B signaling pathway (Signal 1) and the assembly of the NLRP3 inflammasome (Signal 2) (Figure 3). The activation of the NF-*κ*B signaling pathway triggers proinflammatory responses. PAMPs and DAMPs interact with PRRs, leading to the activation of the IkappaB (I*κ*B) kinase (IKK) complex. This activation results in I*κ*B degradation, the release and nuclear translocation of NF-*κ*B, and subsequently increases in the transcriptional levels of NLRP3 and pro-IL-1*β*/pro-IL-18. The NLRP3 inflammasome, a critical cytoplasmic protein complex, encompasses key components including NLRP3, ASC, and caspase 1, and is pivotal in the immune system’s response to pathogenic invasion and cellular stress [54]. When immune cells and barrier epithelium detect invading pathogens or danger signals, these components assemble into a cell pattern recognition receptor called an inflammasome and activate inflammatory cysteine proteases, such as caspases 1, 4, 5, and 11 [55]. During the activation stage, NLRP3 and ASC proteins undergo oligomerization and recruit precursor caspase 1, thereby inducing its self-cleavage into activated caspase 1. The activated caspase 1 further cleaves and activates pro-IL-1*β* and pro-IL-18 [56]. In addition, caspase 1 can also induce pyroptosis, a proinflammatory form of cell death [57]. The caspase 1-dependent pore formation during pyroptosis leads to osmotic analysis of affected host macrophages. Caspase 1 cleaves GSDMD to produce an active N-terminal fragment of GSDMD (GSDMD-N), which binds to acidic phospholipids on the inner leaves of the plasma membrane, oligomerizes to form membrane pores, destroys the cell membrane, and releases inflammatory cytokines and cytokines, including the interleukin 1 (IL-1) family cytokines. Reportedly, mitochondrial dysfunction further activates NLRP3 inflammasome, exacerbating the inflammatory response [58].

### 4.1. Mitochondrial Damage Promotes NLRP3 Inflammasome Activation

Viral infections can trigger immune responses that result in mitochondrial dysfunction in the host cell. Notably, viral infections not only enhance the permeability of the mitochondrial membrane and release of inflammatory factors into the cytoplasm but also induce the translocation of cardiophospholipids from the inner to OMM. It has been shown that ROS, mtDNA, and cardiolipin (CL) translocated to OMM released by mitochondrial dysfunction can trigger the formation and activation of NLRP3 [59,60,61]. Upon DNA binding, cGAS facilitates the synthesis of 2′3′-cGAMP.

During mitochondrial dysfunction, the increase in membrane permeability releases mtDNA, which exacerbates the activation and release of NLRP3 inflammasome and promotes the occurrence of inflammatory responses. The released mtDNA can be recognized by cytosolic cGAS. Upon binding DNA, cGAS and dsDNA form phase-separated condensates in which cGAS catalyzes the synthesis of the cGAS catalyzes the synthesis of 2′3′-cGAMP. cGAMP activates TBK1 through STING, phosphorylates, and activates IRF3, ultimately promoting the expression of IFNs and their downstream target genes (ISGs), thereby initiating antiviral immune responses [62,63,64,65]. Zhong et al. revealed a new mechanism: the TLR signaling activates new mtDNA synthesis through IRF1 and cytidine monophosphate kinase (CMPK2), in which the TLR signaling not only plays a role on the cell surface or cytoplasm but also deeply affects mitochondrial function. Subsequently, the ox-mtDNA is released into the cytoplasm, activating NLRP3 inflammasome and downstream release of IL-1*β* and IL-18, indicating that the ox-mtDNA triggers stronger NLRP3 responses under TLR induction than under TNF stimulation [66].

Mitochondrial ROS (mROS) acts as a signaling molecule to regulate the activation and function of immune cells. They can activate intracellular signaling pathways, and promote the production and release of inflammatory factors, thereby enhancing the antiviral activities of immune cells against pathogens [67]. mROS can also induce the expression of antiviral genes and enhance proinflammatory responses. mROS can promote the binding of the NIMA-related kinase 7 (NEK7) to NLRP3, accelerating inflammasome assembly by binding to quinone oxidoreductase 2 (NQO2) or disrupting the function of mitochondrial complex I. mROS can induce a time-dependent dissociation of the thioredoxin-interacting protein (TXNIP) from thioredoxin (TRX) and then binding to NLRP3, activating NLRP3 inflammasome [68]. Treatment with ROS inhibitors, such as diphenylammonium iodide (DPI) or *N*-acetyl-L-cysteine (NAC), will block the upregulation of NLRP3 transcription [69]. ROS is an important upstream signal for the assembly of NLRP3 inflammasome, which can be activated only when the accumulated mROS in the cell exceeds a certain threshold. However, not all the agonists activate inflammasomes in a ROS-dependent manner.

During mitochondrial dysfunction, translocation of OMM phospholipids CL can activate the NLRP3-mediated pyroptosis. As a key component of OMM, cardiophospholipids usually exist in the form of unsaturated chains in mitochondria, and their distribution is asymmetric, mainly concentrated on the concave surface of mitochondria [70]. This unique molecular structure allows cardiolipin to play a key role in the oxidative phosphorylation of mitochondria and to maintain the structure and function of mitochondria. In the physiological state, the unsaturated nature of cardiolipin makes it susceptible to the induction of ROS, which moves from IMM to OMM, a process that is an important marker of the recognition and clearance of damaging mitochondria by the autophagic system [71]. However, in the presence of severe peroxidative stress, the massive accumulation of cardiolipids on OMM may lead to the recruitment of BAX proteins, which in turn triggers the formation of mPTP, leading to the release of CytC and mtDNA, which are key events in pyroptosis [70,71,72].

The activation of the NLRP3 inflammasome is closely related to the pyroptosis process induced by viral infections. The alteration of the NLRP3 activation state is directly related to exposure to CL in the OMM. During viral infection, cardiolipin exposure to the OMM was associated with enhanced NLRP3 and caspase-1. CL exposure promotes the formation and activation of NLRP3 inflammasome, mediating the occurrence of pyroptosis. OMM cardiolipin may promote the formation of functional inflammasomes by changing its conformational or aggregate state through direct interaction with NLRP3. The formation of functional inflammasomes further activates the cysteine protease caspase-1, leading to the maturation and release of proinflammatory cytokines IL-1*β* and IL-18, ultimately inducing pyroptosis. Migration of CL to OMM can be induced by the activation of NLRP3, while the reduction of CL inhibits the recruitment and activation of NLRP3 [73,74]. Decreased CL synthesis leads to impaired IL-1*β* secretion induced by NLRP3 stimulation and blocked activation of caspase 1. Among these inflammasomes that are exposed to OMM phospholipids, CL can directly bind to caspase 1 and induce oligomerization of caspase 11, but it has no effect on caspase 3 or 8 [75].

### 4.2. Viruses Manipulate the Immune Responses Associated with Mitochondrial Damage

Viral infections usually trigger mitochondrial dysfunction, induce the release of proinflammatory cytokines, and facilitate the elimination of invading viruses in host cells. However, viruses have evolved multiple strategies to manipulate the immune responses triggered by mitochondrial damage, thereby aiding in the evasion of the host immune response. Most of these strategies attenuate the intensity and duration of the proinflammatory responses by reducing the release of proinflammatory cytokines.

MAVS is the main target of viruses that evade host immune responses [76]. The glycoprotein US9 of HCMV can inhibit MAVS. As US9 accumulates in the mitochondria, the mitochondrial membrane potential gradually decreases, which facilitates the leakage of MAVS from the mitochondria [77].

Additionally, the NS3/4A protease of HCV cleaves by targeting MAVS at the Cys-508 site, causing MAVS to lose its function of localization on mitochondria and further inhibiting downstream IFN-*β* induction [78]. Inhibiting the downstream pathway of mtDNA is also a typical immune evasion pathway. STING can activate downstream signaling pathways and induce the production of type I IFNs after detecting intracellular DNA viruses [79]. The E7 protein encoded by the human papillomavirus type 18 (HPV18) can directly bind to STING, inhibit STING function, and block this pathway. The human papillomavirus type 16 (HPV16) E7 protein hijacks the NOD-like receptor X1 (NLRX1) to degrade STING [80]. In addition, viral proteins can also interfere with the apoptotic pathway to allow infected cells to survive for a longer period of time, upregulating the host’s antioxidant mechanism and encoding antioxidant proteins, thereby weakening immune responses and facilitating immune escape. Therefore, evading the immune responses triggered by mitochondrial damage represents a key bioprocess of viral infections.

In addition, viruses have evolved multiple strategies to manipulate autophagy for immunoevasion, while autophagy can be hijacked by viruses for their replication. It has been shown that inflammatory products can modulate autophagy, whereas viruses can induce mROS to trigger autophagy activation and facilitate replication.

Pharmacological intervention or genetic manipulation that inhibits autophagy or lysosomal degradation can increase the effect of mitochondrial DAMPs (mtDAMPs) in the cytoplasm, thereby augmenting the cGAS, inflammasome, and TLR9 signaling pathways [81]. It has been shown that deleting the *pink1* and *prkn* genes (encoding PINK1 and parkin, respectively), can lead to an increase in inflammatory signaling [82]. Furthermore, the activated inflammasome signaling triggered by NF-*κ*B promotes the PRKN-dependent mitochondrial autophagy and clearance of inflammasomes.

Viral infection can impede the normal function of the SNAP29-Stx17-VAMP8 complex, thereby impairing the fusion of autophagosomes with lysosomes, resulting in the accumulation in host cells. Viruses exploit accumulated autophagosomes as sites for replication and assembly, as well as structures for budding. In addition, the PRRSV GP5 protein can induce the mROS/AMP-activated protein kinase (AMPK)/mTOR/ULK1-dependent autophagy, revealing the complexity of the interaction between the virus and host cells [83]. The induction of autophagy by mROS emphasizes the multifaceted nature of virus-host interactions.

Mitochondrial damage induced by autophagy can directly or indirectly affects the host’s immune responses. Autophagy represents a strategy hijacked by viruses for viral replication. Therefore, it is important to explore the interplay among autophagy, viral replication, and immune responses.

## 5. Targeted Regulation of Mitochondrial Damage in Antiviral Therapies

Mitochondria, as the site of oxidative phosphorylation, are the main organelles providing energy for cells. Therefore, viral proteins target mitochondria and alter metabolic pathways to enhance pathogenicity or virulence. During viral infections, PAMPs activate innate immune responses, triggering the production of various proinflammatory cytokines to ensure the host’s ability to resist virus invasion [82,83]. Viral infection leads to mitochondrial internal environment disruption, manifested as increased mitochondrial membrane permeability, oxidative stress, and mtDNA damage, thereby affecting cell viability and immune response. Therefore, the regulation of mitochondrial damage is a cutting-edge and potentially effective approach in the field of antiviral therapy [84,85].

### 5.1. Regulation of Mitochondrial Membrane Permeability in Antiviral Strategies

Viral infection may cause mitochondrial dysfunction, which in turn affects the integrity and permeability of the mitochondrial membrane, such as the changes in the mPTP and mitochondrial membrane potential, as well as electron transport and energy production [83,86,87].

The production of ROS and the release of mtDNA may be associated with the opening of mPTP and VDAC1. The mPTP is an important channel regulating the balance between the internal and external mitochondrial environments. Abnormal opening of mPTP can lead to excessive calcium ion influx into mitochondria, increasing mitochondrial membrane permeability, and resulting in mitochondrial dysfunction and cell apoptosis [88]. Reportedly, the pharmacological inactivation of mPTP with cyclosporin A (CsA) can prevent the TDP43 (Q331K)-mediated mtDNA leakage into the cytoplasm, and that deletion of the gene encoding the mPTP component peptidylprolyl isomerase D (PPID) can improve the TDP43-mediated mtDNA release into the cytoplasm and downstream proinflammatory responses [89]. Therefore, cyclosporin A and similar mPTP inhibitors might become potential drugs for treating mitochondrial damage caused by viral infections, reducing cell damage and proinflammatory responses by maintaining mitochondrial function. It has been shown that CsA can prevent cell death induced by inflammation by blocking the opening of mPTP, activating Fas/FasL, and caspases, and promoting T cell apoptosis [90,91], suggesting that the distinct effects of CsA on T cells and normal cells could be explored as a promising antiviral strategy. Similarly, Lampl et al. showed that infected liver cells undergo mitochondrial membrane permeability transition in the ROS-mediated calcium ion influx, ultimately leading to cell apoptosis, while normal cells do not exhibit this phenomenon [92]. In virus-infected hepatocytes, the TNF-mediated signaling induces calcium release of ER, leading to mitochondrial permeability transition and cell apoptosis, indicating that mitochondrial permeability transition can selectively eliminate the virus-infected hepatocytes.

In addition, vascular biogenesis inhibitor 4 (VBIT-4), an inhibitor of VDAC1 oligomerization, effectively prevents the cytoplasmic accumulation of mtDNA and inflammation in motor neurons derived from induced pluripotent stem cells (iPSCs) obtained from patients with amyotrophic lateral sclerosis (ALS) patients. Additionally, the gene deletion of *VDAC1* inhibits the expression of proinflammatory cytokines, such as TNF, induced by overexpression of TDP43 [11,89]. Furthermore, the increase in MOMP acts as a trigger for cellular or mitochondrial stress, leading to the release of mitochondrial contents. The formation of large pores on the OMM and the inner membrane protrudes results in the release of mitochondrial DAMPs into the cytoplasm. Subsequently, the DAMPs are sensed by various PRRs, leading to the activation of innate immunity [86]. Overall, the mitochondrial damage induced by viral infection results in changes in mitochondrial membrane permeability, facilitating the release of mtDNA into the cytoplasm. The released mtDNA can be recognized by PRRs and the subsequent activation of downstream signaling pathways, such as cGAS-STING, inducing the production of proinflammatory cytokines. Therefore, targeting regulators of mitochondrial membrane permeability could be an important strategy in reducing mitochondrial damage and preventing mitochondrial DNA leakage during viral infections.

### 5.2. Modulation of the Mitophagy Pathway in Response to Viral Infections

Mitophagy, an intracellular self-clearance and repair mechanism, plays a crucial role in maintaining cellular homeostasis and responding to external stress. During viral infections, changes in mitochondrial membrane permeability and ROS generation lead to mitochondrial damage, resulting in the leakage of mitochondrial components into the cytoplasm [93]. Mitophagy maintains intracellular homeostasis by eliminating damaged and dysfunctional mitochondria. Selective mitophagy requires regulation through autophagy receptors/adaptor proteins, after which mitochondria are engulfed into double-membrane structures called autophagosomes, ultimately leading to organelle degradation upon fusion with lysosomes [94].

Mammalian mitophagy can be classified into two distinct pathways: the PINK1-dependent and PRKN-independent pathways. It has been shown that various viruses, including HBV, HCV, classical swine fever virus, and coxsackievirus B, can induce mitophagy through the activation of the pink1-arkin pathway. During viral infection, phosphorylation of DNM1L (Ser616) is upregulated, promoting DNM1L recruitment to mitochondria and leading to mitochondrial fission, which initiates the PRKN-dependent mitophagy [93,94,95,96,97,98]. Therefore, drugs targeting Drp1 offer the potential for exploring new antiviral strategies, such as mitochondrial division inhibitor 1 (mdivi-1).

Mitophagy can be induced via the PRKN-independent pathways in the absence of PINK1 or Parkin protein in eukaryotic cells. Viruses, such as severe acute respiratory syndrome coronavirus (SARS-CoV), human herpesvirus 8 (HHV-8), and human parainfluenza virus 3 (HPIV3), promote mitochondrial recruitment to autophagosomes through interactions between autophagy receptors containing typical LIR motifs and LC3 [93,94,99]. Overall, selectively targeting cellular autophagy pathways may be a potential therapeutic strategy for antiviral treatment.

Although mitophagy is an important step in enhancing the host cell’s ability to inhibit viral replication, some viruses can manipulate mitophagy through various molecular mechanisms to facilitate immunoevasion and enhance virus replication. The related mechanisms mainly include viruses using mitophagy to inhibit the signaling pathways of type I IFNs, inflammation, and apoptosis [93,94].

Viruses have evolved several strategies to induce mitophagy and inhibit type I IFN production. For example, the hepatitis B virus induces the parkin-dependent mitophagy to recruit the linear ubiquitin assembly complex (LUBAC) to mitochondria, disrupting the formation of MAVS signaling complexes and the subsequent induction of type I IFNs. The SARS-CoV ORF9b degrades MAVS by hijacking the poly(rC)-binding protein 2 (PCBP2) and ITCH E3 ubiquitin protein ligase (ITCH/AIP4) [100].

Damaged mitochondria can release DAMPs that contribute to the activation of the NLRP3 inflammasome, thereby triggering the secretion of proinflammatory cytokines necessary for innate immunity and the activation of adaptive immune responses [101]. However, viral proteins can also inhibit the activation of the NLRP3 inflammasome, facilitating immunoevasion. It has been shown that measles virus (MeV) infection triggers the release of IL-1*β* through the NLRP3 inflammasome in THP-1 cells. However, the MeV non-structural V protein inhibits the process, reducing IL-1*β* production [93,94,102].

Selective intervention in cellular autophagy pathways can regulate mitochondrial damage and play an important role in antiviral therapy [103]. Autophagy pathway modulators, including autophagy inhibitors (such as chloroquine and 3-methyladenine) and autophagy promoters (such as rapamycin and spermidine), can regulate cellular autophagy pathways, thereby impairing the process of mitophagy, which in turn affects viral replication and the immune response capabilities of host cells [104,105].

## 6. Conclusions and Perspectives

In this review, we summarize the involvement of mitochondrial dysfunction in viral infections, emphasizing its regulatory role in innate immune responses through energy metabolism, redox balance, and cellular signaling pathways. We discuss the effects of mROS on host immune responses during viral infection, thereby regulating viral replication in host cells. Furthermore, we underscore the pivotal roles of the mitochondrial membrane-associated components in the activation of proinflammatory mediators and cellular pyroptosis.

Notably, emerging evidence suggests an intricate crosstalk between mitochondria dysfunction and immune responses, wherein the same molecule may exert diverse effects on viral replication, either inhibiting or promoting it. The interplay between the cellular proteins associated with mitochondrial damage and immune responses constitutes a multifaceted and self-regulating process. Despite having outlined the interactions, the molecular mechanisms need further investigation.

Future studies are required to investigate the regulatory effects of mitochondrial damage-related factors on immune responses as well as the precise mechanisms of these regulatory pathways. Additionally, antiviral treatment strategies targeting mitochondrial damage have the potential to alleviate cell damage and proinflammatory responses induced by viral infections.

## Figures and Tables

**Figure 1 ijms-25-09206-f001:**
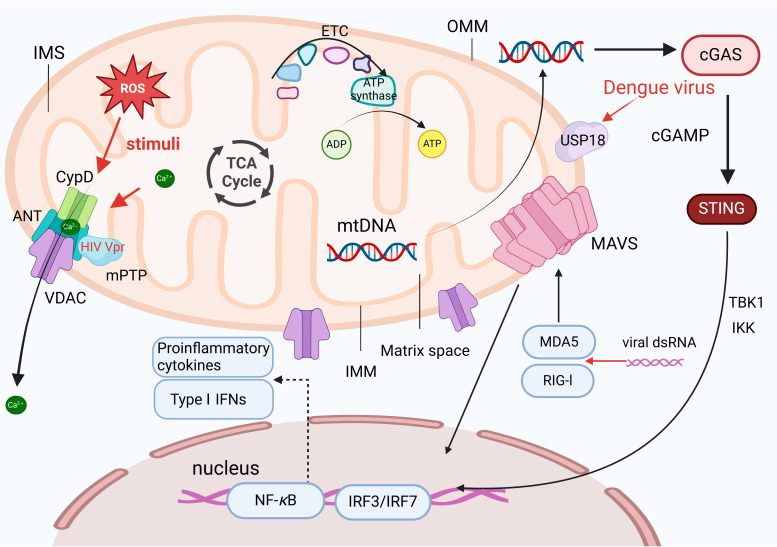
Mitochondria are essential organelles with a double-membrane structure, consisting of the outer membrane (OMM), intermembrane space (IMS), inner membrane (IMM), and matrix. The IMM folds into cristae, which increase the surface area for the electron transport chain (ETC) and ATP synthase, facilitating oxidative phosphorylation. Viral infections, such as HIV, HCMV, and HCV, can disrupt mitochondrial membranes. The HIV Vpr protein can directly bind to adenine nucleotide translocator (ANT), promoting the opening of the mitochondrial permeability transition pore (mPTP), and leading to the release of mitochondrial DNA (mtDNA), which acts as a danger signal, activating pattern recognition receptors (PRRs) and triggering inflammatory responses. The mitochondrial antiviral-signaling protein (MAVS) on the outer mitochondrial membrane mediates RIG-I and MDA5 signaling, activating the IRF3 and nuclear factor kappa B (NF-*κ*B) pathways to enhance the expression of type I interferons (IFNs), such as IFN-*β*, driving the antiviral response. Dengue virus indirectly promotes mtDNA release by upregulating USP18, exacerbating inflammatory responses.

**Figure 2 ijms-25-09206-f002:**
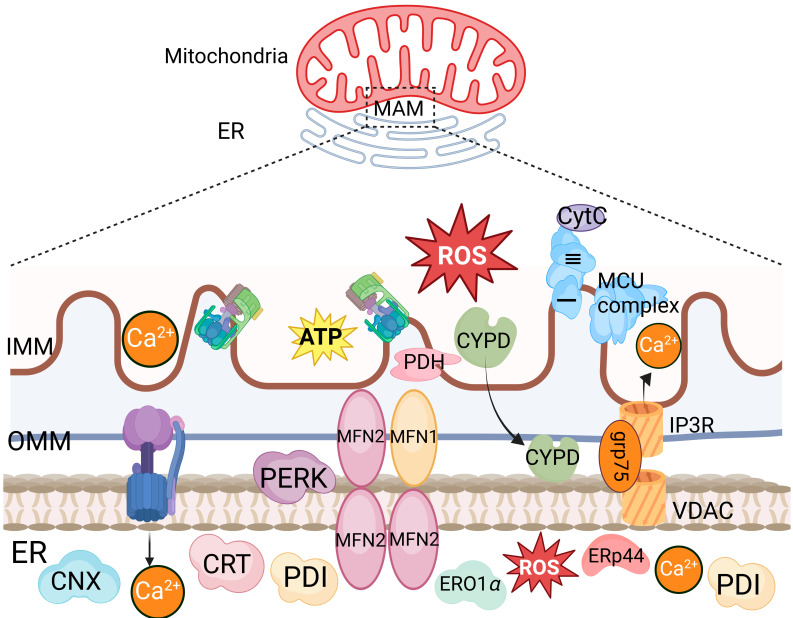
The structure of IP3R-GRP75-VDAC1 complex. An enlarged image of the structure of IP3R-GRP75-VDAC1 complex, which illustrates the interaction between ER and mitochondria, characterized by the formation of key protein complexes, including inositol 1,4,5-trisphosphate receptor (IP3R), voltage-dependent anion-selective channel (VDAC), protein disulfide isomerase (PDI), ER-resident protein 44 (ERp44), and mitofusin 1 (MFN1). The complex engages in calcium ion transfer, signal transduction, and energy metabolism.

**Figure 3 ijms-25-09206-f003:**
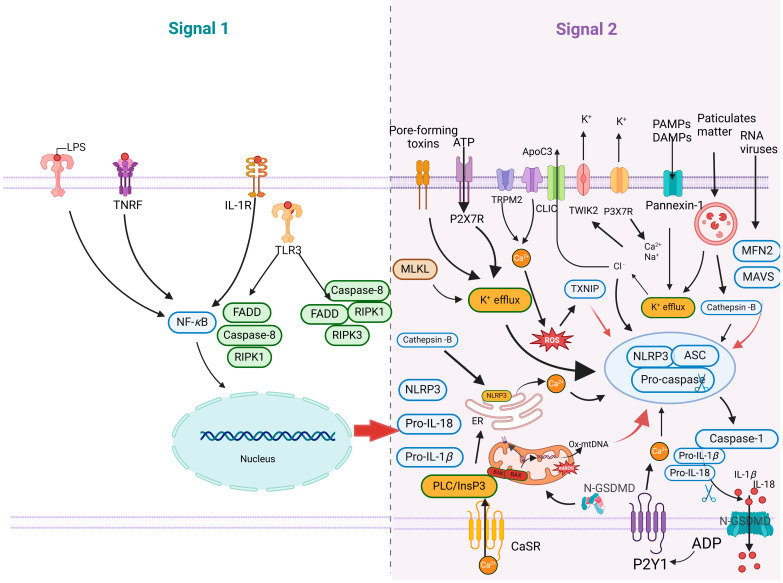
Schematic diagram of the canonical NLRP3 inflammasome activation. Signal 1 (the priming stage, left panel) is induced by Toll-like receptors (TLRs), nucleotide-binding oligomeric structural domain protein (NODs), and TNF receptor (TNFR), which recognize pathogen-associated molecular patterns (PAMPs) or damage-associated molecular patterns (DAMPs), leading to the upregulation of NOD-like receptor thermal protein domain associated protein 3 (NLRP3) and the proinflammatory cytokines interleukin 1*β* (IL-1*β*) and interleukin 18 (IL-18). Signal 2 (the activation stage, right panel) involves the oligomerization of NLRP3 and the assembly of the NLRP3, ASC, and pro-caspase-1 complex, which is typically triggered by ATP, pore-forming toxins, viral RNA, and particulate matter. Consequently, cellular signaling events such as K^+^ efflux, Ca^2+^ influx, ROS production, mitochondrial dysfunction, lysosomal rupture, and chloride intracellular channel-dependent Cl^−^ efflux can be initiated. The formation of the NLRP3 inflammasome activates caspase-1, which subsequently cleaves pro-IL-1*β*/pro-IL-18 into IL-1*β*/IL-18. Additionally, GSDMD is cleaved by caspase-1 and inserted into the membrane, causing pore formation and pyroptosis.

**Table 1 ijms-25-09206-t001:** Characteristics of virus-induced mitochondrial damage.

Types of Mitochondrial Damage	Viruses/Viral Proteins	Targeted Pathways/Functional Roles	References
Increased mitochondrial membrane permeability	PDCoV	BAX-mediated MOMP	[13]
	HSV-1 and SFV	Triggering MOMP through the Puma protein	[14]
	PRRSV GP5	Activation of IP3R-GRP75-VDAC1	[15]
	DENV	Release of the proinflammatory cytokines-induced mtDNA	[16]
Disruption of mitochondrial dynamics	ZIKV	Reduction of the MFN2 protein	[17]
	HIV Vpr	Reduction of the Vpr-related MFN2	[18]
	Influenza virus NS1	The fragmentation of mitochondria	[19]
	HBV/HCV	PRKN-dependent mitochondrial autophagy	[20,21]
	IAV	Regulation of the autophagy-related signaling pathways	[22]
Mitochondrial stress	HCV E1, E2, and NS3	Inhibition of electron transfer	[23]
	HIV Tat	The Tat protein is translocated from the nucleus to the mitochondria	[24]

## Data Availability

Data supporting the reported results are available in this article.

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
