# Peer review of "Crosstalk between Dysfunctional Mitochondria and Proinflammatory Responses during Viral Infections"

_ijms, 2024, doi:10.3390/ijms25179206_

Round 1

Reviewer 1 Report

Comments and Suggestions for Authors

In the paper entitled "Crosstalk Between Dysfunctional Mitochondria and 2 Pro-inflammatory Responses During Viral Infections", the authors describe in detail the mechanisms involving mitochondria during viral infections. In particular, the authors describe the role of mitochondria in inflammation during viral infection and the mechanisms of mitochondrial damage as targets for antiviral therapies .

The topic is relevant to understanding the role of mitochondria in viral infections.
The review is well organized  and I think it will be of interest to the readers of the journal and it may be helpful to researchers  that investigate the mechanisms of viral infection of mammalian cells.

Manuscript should be accepted with minor revisions

Minor revisions:

The authors use many abbreviations in the text. To help the readers a list of the most important abbreviations of eznymes should be included in the text.

The data on each type of virus described in the manuscript and the mitochondrial proteins and pathways involved in the infection mechanisms should be summarized in one or more tables.

Figure 3: The author should better explain the meaning of "Signal 1" and "Signal 2" in the figure caption and clearly show the connection between the Signal 1 and Signal 2 diagrams.

Author Response

Reviewer #1

Comments and Suggestions for Authors

In the paper entitled "Crosstalk Between Dysfunctional Mitochondria and 2 Pro-inflammatory Responses During Viral Infections", the authors describe in detail the mechanisms involving mitochondria during viral infections. In particular, the authors describe the role of mitochondria in inflammation during viral infection and the mechanisms of mitochondrial damage as targets for antiviral therapies .

The topic is relevant to understanding the role of mitochondria in viral infections.
The review is well organized, and I think it will be of interest to the readers of the journal and it may be helpful to researchers  that investigate the mechanisms of viral infection of mammalian cells.

Authors’ response: We are grateful for the opportunity to revise and improve our manuscript. We greatly appreciate the positive comments and valuable suggestions. The point-by-point responses to the comments of all the reviewers are provided below. All revisions are highlighted in red in the revised manuscript.

Manuscript should be accepted with minor revisions

The authors use many abbreviations in the text. To help the readers a list of the most important abbreviations of eznymes should be included in the text.
Authors’ response: The list of abbreviations has been included in the revised manuscript (Lines 599–614).

The data on each type of virus described in the manuscript and the mitochondrial proteins and pathways involved in the infection mechanisms should be summarized in one or more tables.

Authors’ response: We have summarized a table about the involvement of mitochondrial proteins or pathways on the viral infection mechanisms in the revised manuscript (Table 1).

Figure 3: The author should better explain the meaning of "Signal 1" and "Signal 2" in the figure caption and clearly show the connection between the Signal 1 and Signal 2 diagrams.

Authors’ response: Excellent suggestions. We have provided the meanings of "Signal 1" and "Signal 2" in the revised manuscript (Lines 302–314). In addition, the connection between “Signal 1" and "Signal 2” has been included in the updated Figure 3.

Reviewer 2 Report

Comments and Suggestions for Authors

The review article submitted by Sun et al entitled “Crosstalk Between Dysfunctional Mitochondria and Pro-inflammatory Responses During Viral Infections” to IJMS provide evidence for the interrelation between mitochondrial (Mt) function, its damage and pro inflammatory cytokine release during viral infection. The authors have provided four sections in this review (i) Mt structure and functions (ii) Mt damage induced by viral infections (iii) Proinflammatory response induced by mt damage and finally (iv) Targeted regulation of mt damage. I appreciate the author’s effort in cumulating data supporting the proinflammatory response and mt damage. However, the article misses orientation or not precise to the points discussed. I would encourage authors to include more virus data in each section and expand all sections. Also the authors need to convert the sentences to be more concise to the point. Below are my few major comments which will help to improve further.

(i) Mitochondrial Structure and Functions: Mitochondria is the most complicated organelle having very critical functions. This section currently provided looks very meagre or marginal. Functions were not discussed well. Kindly expand and additionally include evidence for mt protein interacting with viruses/or enzymes modulated / mt functions altered in terms of viral infection. The figure provided Figure 1 doesn’t add any new information. Rather modify the figure with above mentioned data and show which viruses act on different region of mitochondria. When you expand this section, the information provided on 3.1 about the mPTP, IMM and OMM can be moved to here. Thus, everything related to structure and composition of Mt can be detailed under this section.

(ii) Figure 2: The crosstalk among organelles doesn’t fit in the current section. This can be a separate section after the structure and function describing main pathways involved. Figure 2a is not required. Rather show a small inset on 2b the crosstalk of ER and Mt. Kindly expand this section well.

(iii) Mt damage induced by viral infections-It would be good to reorient this section in terms of signaling /metabolic pathways described, rather than the current titles.

(iv) A general introduction of various proinflammatory cytokine signaling pathways during viral infection in the beginning will help to avoid the repetition of several pathways in main sections. This will help to focus on the mt role in section iii) Proinflammatory response induced by mt damage

(v) Several sentences provide same idea in different scenario. This kind of sentences can be avoided and try to make the article very concise to the point discussed.

Author Response

Reviewer #2

Comments and Suggestions for Authors

The review article submitted by Sun et al entitled “Crosstalk Between Dysfunctional Mitochondria and Pro-inflammatory Responses During Viral Infections” to IJMS provide evidence for the interrelation between mitochondrial (Mt) function, its damage and pro inflammatory cytokine release during viral infection. The authors have provided four sections in this review (i) Mt structure and functions (ii) Mt damage induced by viral infections (iii) Proinflammatory response induced by mt damage and finally (iv) Targeted regulation of mt damage. I appreciate the author’s effort in cumulating data supporting the proinflammatory response and mt damage. However, the article misses orientation or not precise to the points discussed. I would encourage authors to include more virus data in each section and expand all sections. Also the authors need to convert the sentences to be more concise to the point. Below are my few major comments which will help to improve further.

Author’s response: Thanks for your professional suggestions. We have enriched the data on viral infections as suggested. Furthermore, we have revised several sentences for conciseness and clarify.

  • Mitochondrial Structure and Functions: Mitochondria is the most complicated organelle having very critical functions. This section currently provided looks very meagre or marginal. Functions were not discussed well. Kindly expand and additionally include evidence for mt protein interacting with viruses/or enzymes modulated / mt functions altered in terms of viral infection. The figure provided Figure 1 doesn’t add any new information. Rather modify the figure with above mentioned data and show which viruses act on different region of mitochondria. When you expand this section, the information provided on 3.1 about the mPTP, IMM and OMM can be moved to here. Thus, everything related to structure and composition of Mt can be detailed under this section.

Author’s response: As suggested, we have added some information, including the interactions between viral proteins and mitochondrial proteins, Ca2+ release, and the crosstalk between mitochondria and innate immune signaling pathways, etc., in the updated Figure 1. Additionally, the information about the mPTP, IMM, and OMM has been moved to the section.

Figure 2: The crosstalk among organelles doesn’t fit in the current section. This can be a separate section after the structure and function describing main pathways involved. Figure 2a is not required. Rather show a small inset on 2b the crosstalk of ER and Mt. Kindly expand this section well.

Author’s response: Thanks for your valuable suggestions. The crosstalk among various organelles (Figure 2a) has been removed. The description of crosstalk among organelles cannot serve as a separate section due to the limited association with the theme. Furthermore, the crosstalk of ER and Mt has been expanded as you suggested (updated Figure 2).

  • Mt damage induced by viral infections-It would be good to reorient this section in terms of signaling /metabolic pathways described, rather than the current titles.

Author’s response: This section presents diverse forms of mitochondrial damage induced by viral infections, which cannot be succinctly encapsulated in a title about signaling/metabolic pathways. Furthermore, the original title and the subsequent section's focus on “Proinflammatory responses induced by mitochondrial damage upon viral infections” exhibit logical progression, and the modification would disrupt the coherence of both sections. Hence, we retained the original title.

  • A general introduction of various proinflammatory cytokine signaling pathways during viral infection in the beginning will help to avoid the repetition of several pathways in main sections. This will help to focus on the mt role in section iii) Proinflammatory response induced by mt damage

Author’s response: As suggested, we have moved the primary proinflammatory response signaling pathways, including the NF-κB and inflammasome signaling, to the beginning of Section 4.

  • Several sentences provide same idea in different scenario. This kind of sentences can be avoided and try to make the article very concise to the point discussed.

Author’s response: We have revised the manuscript and deleted several sentences with the same idea in different scenarios.

Comments 2: Figure 2: The crosstalk among organelles doesn’t fit in the current section. This can be a separate section after the structure and function describing main pathways involved. Figure 2a is not required. Rather show a small inset on 2b the crosstalk of ER and Mt. Kindly expand this section well.

Response 2: Thanks for your valuable suggestions. The crosstalk among various organelles (Figure 2a) has been removed. The description of crosstalk among organelles cannot serve as a separate section due to the limited association with the theme. Furthermore, the crosstalk of ER and Mt has been expanded as you suggested (updated Figure 2).

Comments 3: Mt damage induced by viral infections-It would be good to reorient this section in terms of signaling /metabolic pathways described, rather than the current titles.

Response 3: This section presents diverse forms of mitochondrial damage induced by viral infection, which cannot be succinctly encapsulated in a title about signaling/metabolic pathways. Furthermore, the original title and the subsequent section's focus on “Pro-inflammatory response induced by mitochondrial damage upon viral infections” exhibit logical progression, and the modification would disrupt the coherence of both sections. Hence, we retained the original title.

Comments 4: A general introduction of various proinflammatory cytokine signaling pathways during viral infection in the beginning will help to avoid the repetition of several pathways in main sections. This will help to focus on the mt role in section iii) Proinflammatory response induced by mt damage

Response 4: As suggested, we have moved the primary proinflammatory response signaling pathways, including the NF-κB and inflammasome signaling, to the beginning of the Section 4.

Comments 5: Several sentences provide same idea in different scenario. This kind of sentences can be avoided and try to make the article very concise to the point discussed.

Response 5: We have revised the manuscript and deleted several sentences with same idea in different scenario.

Round 2

Reviewer 2 Report

Comments and Suggestions for Authors

The authors have addressed my concerns and issues raised before and the review article is organized very nicely. New figure has been incorporated. Please check for the cosmetic errors as well as the abbreviations used. Few Bcl-2 family proteins were written in capital letters. It needs to be corrected. Otherwise looks good.